# Microbial mats promote surface water retention in proglacial streams

Jonas Paccolat[1], Pietro de Anna[2], Stuart Lane[3], Hannes Peter[1], and Tom Battin[1]

[1]River Ecosystems Laboratory, École Polytechnique Fédérale de Lausanne (EPFL), Sion, Switzerland
[2]Institute of Earth Sciences, Université de Lausanne, Lausanne, Switzerland
[3]Institute of Earth Surface Dynamics, Université de Lausanne, Lausanne, Switzerland

**Correspondence:** Tom Battin (tom.battin@epfl.ch)

**Abstract.** The retreat of glaciers opens up large proglacial areas which become available for colonization and primary succession. Yet, factors that contribute to habitability during early succession in proglacial areas remain poorly understood. In proglacial streams, biofilms, which are matrix-enclosed microbial communities, colonize the streambed and grow into millimeter thick mats. Particularly in proglacial streams draining relatively flat and stable lateral terraces, these biofilms may augment habitability by reducing water scarcity through clogging of the streambed. To quantitatively address this phenomenon, we performed streamside flume experiments and conceived the idealized *terrace model*, which models stream length elongation as a function of microbially induced clogging, sediment hydraulic properties, stream roughness, slope, width and inflow. Significant stream elongation, and hence habitabilization, occurs when clogging suffices to induce unsaturated conditions below the streambed. Considering multiple terrace configurations with educated parameter bounds, we found a wide range of possible elongation, ranging from none to 100-fold. Sensitivity analysis suggests that sediment hydraulic properties mostly contribute to variability in stream elongation due to biofilm induced clogging. Taken together, we here show that microbial communities can significantly extend the habitability of proglacial stream ecosystems by inducing streambed clogging and retaining water. This is relevant in light of the rapid glacier retreat.

## 1 Introduction

Globally, glaciers are projected to lose $20\,\%$ to $60\,\%$ of their actual surface by the end of the century exposing large areas to colonization and primary succession (Bosson et al., 2023). Understanding the formation of newly deglaciated ecosystems is critical for anticipating consequences for their ecohydrology and ontogeny (e.g., Gurnell et al., 2000; Eichel, 2019). It has been postulated that geomorphological and ecological processes in proglacial terrain were mostly independent and that primary succession progressed linearly with time since deglaciation (Cooper, 1939; Matthews and Whittaker, 1987). This notion has recently been redressed, acknowledging the coupling of geomorphological and ecological processes (Ficetola et al., 2021). It is well established that disturbance regimes, soil moisture, sediment size and microclimate, for instance, affect the distribution

of species within glacier forefields (e.g., Siegfried et al., 2023; Burga et al., 2010; Roncoroni et al., 2023b). However, there is also building evidence that certain organisms can augment ecosystem habitability by altering the physical environment (Eichel, 2019; Miller and Lane, 2019; Roncoroni et al., 2019). Miller and Lane (2019) synthesized previous models (from Matthews (1992) and Corenblit et al. (2007)) into a theoretical framework of primary succession accounting for environmental stress and the balance between biotic and abiotic controls in glacier forefields (Fig. 1a). This model identifies four ecosystem phases: (1) in the geomorphic phase, abiotic factors preclude life settlement; (2) in the pioneer phase, abiotic factors dominate and limit life to pioneer species; (3) in the biogeomorphic phase, biotic and abiotic factors contribute equally; and finally (4) in the ecological phase biotic factors dominate. Depending on severity and frequency of environmental stress, primary succession can remain limited to the geomorphic phase, or reset to the pioneer phase. Under limited stress, primary succession in proglacial landscapes can transit towards biogeomorphic and ecological controls.

Following incision–aggradation mechanisms (Maizels, 2002), terraces often form between the active floodplain and lateral moraines (both typical elevated stress environments) and offer relatively stable terrain compatible with primary succession. Surface water retention is very limited on these terraces making water scarcity the main stress. Nascent soils are characterized by high permeability as they are composed of poorly sorted sediments of heterogeneous sizes (Burga et al., 2010). Furthermore, the elevated location of terraces prevents capillary water rise (Siegfried et al., 2023). High permeability coupled with disconnection results in large infiltration rates (Brunner et al., 2009), up to of a few meters per hour (Müller et al., 2022), strongly limiting surface water extent. Consequently, water accessibility exerts a strong control on spatial distribution of vegetation. For instance, Siegfried et al. (2023) found that drought tolerant pioneer species could colonize the whole terrace albeit sparsely, while early successional plant species were confined to the vicinity of surface water or to shallow water table locations.

Besides ponding water following rain events, surface water is mainly found within both groundwater-fed and glacier-fed channels (Robinson et al., 2015) (Fig. 1b), which can cover up to $10\%$ of proglacial terrace areas (e.g., Roncoroni et al., 2023b). Intermittency (that is, zero-flow days) is inherent to these terrace streams (i.e., draining proglacial terraces) with expansion–contraction cycles arising from the superposition of daily and seasonal melting cycles and stochastic rain events (e.g., Paillex et al., 2020; Malard et al., 1999; Llanos-Paez et al., 2025). In addition to the diversity of water sources, the heterogeneity of water storage and transmission times within different proglacial landforms contribute to the complexity of the spatiotemporal dynamics of these terrace streams (Langston et al., 2011).

It was recently suggested that microbial mats (or biofilms) coating the streambed may promote the habitability of these streams via fertilization, stabilization and impermeabilization (Miller and Lane, 2019; Roncoroni et al., 2019). After snow melt, microbial mats rapidly grow up to a few millimeters in thickness and accumulate organic matter and nutrients on an otherwise mineral streambed. This fertilization effect benefits biofilm development (Vincent and Howard-Williams, 1986), but also pioneering plants adjacent to the streams (Ciccazzo et al., 2016; Frey et al., 2013). Furthermore, microbial mats are rich in extracellular substances which increase the cohesiveness of fine sediments (e.g., Gerbersdorf and Wieprecht, 2015).

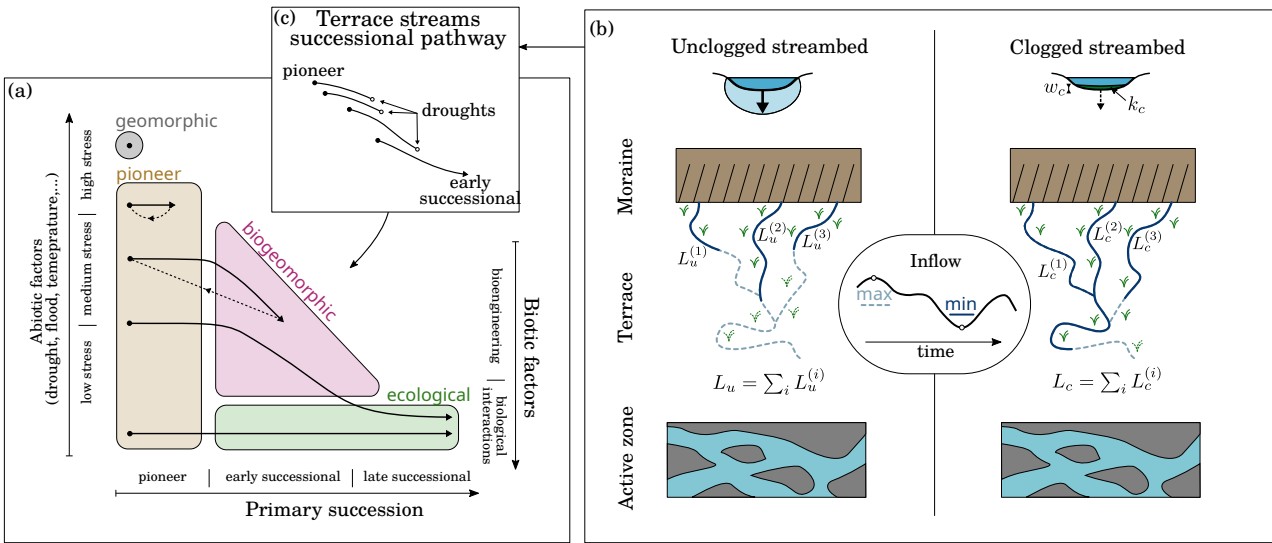

**Figure 1.** (a) Miller and Lane (2019) succession model within proglacial forefields. Ecosystem phases (colored regions) are positioned relative to the importance of (bio-)geomorphic and ecological controls. Different succession pathways (solid black arrows) correspond to different environmental stress. Dashed black arrows indicate disturbances. (b) Conceptual representation of the effect of microbial mats on terrace streams. Expansion–contraction cycles are illustrated by the solid and dashed stream lines, which correspond to minimal and maximal inflows for bio-clogged and unclogged streams, respectively. (c) Terrace stream succession pathway.

Therefore, benthic microbial mats are thought to contribute to the stability of these streams, particularly in the face of flow fluctuations. Finally, microorganisms can contribute to streambed clogging through active growth but also trapping of fine
60  sediments (Dubuis and De Cesare, 2023). Because of water scarcity caused by highly permeable sediments, clogging by microbial mats may extend the wetted perimeter of streambeds and even flow length, therefore expanding their spatial cover. For instance, experimental work showed that the growth of microbial mats significantly reduced infiltration in flumes that mimicked a small proglacial tributary stream (Roncoroni et al., 2023a). Because the impermeabilization effect strongly depends on the substrate, resolving the permeability of clogging layers is needed to predict surface water extension following biofilm
65  development. However the permeability of microbial mats in streams remains poorly studied.

The aim of this study was to quantify potential effect of microbial mats on the linear extension of streams draining proglacial terraces. We asked the following questions: (1) What is the microbial mat permeability and does it vary spatially and temporally? In particular, how is it affected by intermittent desiccation ? (2) Is the permeability small enough to extend the flow length
70  of terrace streams ? If so, what would be the effect of the underlying sediments? (3) Can we quantify surface water extension at the scale of the terrace despite the complexity of intermittent inflows? To address these questions, we first carried out a series of flume experiments in two different Alpine catchments to monitor microbial mat growth and permeability. We also studied

how permeability responds to mat desiccation under controlled experiments. Finally, we conceptualized a simplified model (referred to as *terrace model*, hereafter) to estimate clogging effect on surface water retention. Experimental results are used as input parameters in the terrace model.

## 2  Methods

### 2.1  Experiments

A streamside flume (1 m long) was designed to grow microbial mats under near natural conditions and monitor mat permeability, $k_c$, non-destructively. A middle section of the flume is drained by a box of side length $L = 20$ cm filled with sediments. As streamwater was fed into the flume, microbial mats started colonizing the sediment interface. The mat thickness, $w_c$, was measured with a micrometer screw gauge (0.1 mm accuracy) by averaging over 15 points regularly covering the flume. Monitoring the infiltration rate $q$ and the pressure profile throughout the box (with two columns of nine piezometers each), the mat hydraulic conductivity, $K_c$, and the sediment hydraulic conductivity, $K_s$, are obtained by solving the double Darcy equation

$$q = K_c \left( 1 + \frac{h^{\mathrm{up}} - h_I}{w_c} \right) = K_s \left( 1 + \frac{h_I - h^{\mathrm{dn}}}{L} \right), \tag{1}$$

where $h^{\mathrm{up}}$ is the water depth in the flume (above the box) and $h^{\mathrm{dn}}$ the water column imposed at the box outlet (Fig. 2b). Assuming uniformity of both sediments and mat, an algorithm was developed to find the best value of the interface pressure head $h_I$ (between mat and sediments) with conservative uncertainty range based on the piezometer data (see SI Section 2). Finally, to cancel out the effect of temperature on hydraulic conductivity, $K$ – water viscosity almost doubles in the range of temperature considered –, it is converted to permeability, $k$, according to the monitored temperature. Five experimental runs were carried out: *v1* (17 days) and *v2* (34 days) in an alpine catchment (Valsorey, Switzerland) and *r1* (9 days), *r2* (19 days), and *r3* (20 days) in a subalpine catchment (Réchy, Switzerland). Sediments ranging from 0.1 to 1 mm (fine to coarse sand) were used for *v1*, *v2*, *r1*, and *r2*, while sediments ranging from 1 to 2 mm (very coarse sand) were used for *r3*. The two categories are referred to as fine and coarse sediments, respectively. The flow velocity above the sediments was maintained close to $1.5 \, \mathrm{cm \, s^{-1}}$ during the different runs.

How mat permeability responded to desiccation was investigated on nine sediment cores (10 to 15 cm, 3 cm $\varnothing$) sampled at the end of *r2*. In each core, a millimeter thick mat overlaid the sediment column. After being re-saturated, the cores were dried (30 °C) and triplicates were withdrawn during phase I of evaporation (after 4 days), three more at the transition to phase II (after 15 days), and the last three during phase II (after 25 days) (see SI Section 3 or Lehmann et al. (2008) for a definition of evaporation phases). Immediately after withdrawal, the cores were re-saturated and the column (sediment + mat) permeability, $k(t)$, was measured during eight hours using the falling head method (Klute and Dirksen, 1986). Because the saturation procedure was applied on wet sediments, saturation was incomplete and air entrapment altered permeability. A linear drift, $k_{\mathrm{sed}}(t) = 47(8) \times 10^{-6} + 0.6t \, \mathrm{mm}^2$, was observed on a series of barren sediment cores (see SI Section 3). The

evolution of the mat resistivity, $r_c = w_c/k_c$, after desiccation is thus given by

$$r_c(t) = \frac{l}{k(t)} - \frac{l}{k_{\text{sed}}(t)},$$ (2)

where $l$ is the core height.

## 2.2 Terrace model

The effect of clogging on surface water retention in terrace streams was investigated with the following idealized *terrace model*. The terrace is assumed infinitely long such that streams eventually fade out because of infiltration. Each channel is assumed uniform in width, $W$, slope, $s$, and roughness, $n_M$ (Manning coefficient). To account for the variability of topographic and pedologic conditions, the following (educated) bounds are used: $0.5\,\text{m} < W < 2\,\text{m}$, $10^{-4} < s < 10^{-1}$ and $0.1\,\text{s}\,\text{m}^{-1/3} < n_M < 0.3\,\text{s}\,\text{m}^{-1/3}$. They are chosen to reflect the variability of proglacial streams. Because no specific prior knowledge is available, we consider that channel width and Manning roughness follow uniform distributions; whereas to recognize the important fraction of flat conditions, the slope is assumed log-uniformly distributed. The hydraulic properties of the sediments underlying a channel are characterized by their hydraulic conductivity, $K_a$, and the van Genuchten–Mualem (vG–M) shape and scale parameters, $n$ and $h_g$ (van Genuchten, 1980). Two distinct joint probability distribution are considered: the *sand* distribution of Carsel and Parrish (1988) and the *floodplain* distribution. The latter is introduced because the reported hydraulic conductivity values in glacial floodplains (Müller et al., 2022) are mainly above the upper range of the sand distribution. For the floodplain distribution, $K_a$ is taken log-uniformly between $1 \times 10^{-4}\,\text{m}\,\text{s}^{-1}$ and $8 \times 10^{-3}\,\text{m}\,\text{s}^{-1}$ (i.e., range reported in Müller et al. (2022)) and the associated vG–M parameters are computed according to the model of Peche et al. (2024).

### 2.2.1 Stream length

For a constant discharge $W\,Q_{\text{in}}$ (in $\text{m}^3\,\text{s}^{-1}$), the stream length is given by

$$L = \int\limits_0^{Q_{\text{in}}} \frac{dQ}{q\left(h^{\text{up}}(Q)\right)},$$ (3)

where $q$ is the infiltration rate (in $\text{m}\,\text{s}^{-1}$) which depends on the ponding depth $h^{\text{up}}$ itself found from the Manning rating curve $Q(h^{\text{up}}) = h^{\text{up}} R^{2/3} \sqrt{s}/n_M$, with $R = h^{\text{up}} W/(2h^{\text{up}} + W)$ the hydraulic radius of a rectangular channel. The stream inflow condition is parametrized by the inlet stage, $h_{\text{in}}$, taken randomly between 0 and $0.3\,\text{m}$. The above integral solution relies on the assumption of steady-state conditions: the inflow must be constant during a period longer than the time required for the surface front to propagate to the stream end (where it fades out) and the transient time of infiltration. It is also assumed that the tributary streams are disconnected from the groundwater, yielding an infiltration rate independent of the water table position (Brunner et al., 2009). Note that transient groundwater movement, such as mounding, is admissible as long as it does not alter stream disconnection.

The infiltration rate through a channel uniformly clogged by a layer of thickness, $w_c$, and hydraulic conductivity, $K_c$, is given by the one-dimensional equation (Fox and Durnford, 2003)

$$q_c = K_c \left( 1 + \frac{h^{\mathrm{up}} - h_I}{w_c} \right) = K_a(-h_I), \tag{4}$$

in case of unsaturated streambed and by

$$q_c = K_c \left( 1 + \frac{h^{\mathrm{up}} - h_I}{w_c} \right) = q_u(h_I), \tag{5}$$

in case of saturated streambed, where $h_I$ is the pressure head at the interface between both layers (negative for Eq. (4) and positive for Eq. (5)) and $K_a(\cdot)$ is the unsaturated hydraulic conductivity of the terrace sediments. The unsaturated one-dimensional infiltration (Eq. (4)) is a good approximation as long as $h^{\mathrm{up}} < h_c = w_c(K_c/K_a - 1)$. For larger ponding depths (or equivalently weaker clogging), an inverted water table is formed beneath the stream (Brunner et al., 2009; Xian et al., 2017) and lateral flow must be considered. The equation $q_u(h^{\mathrm{up}}) = K_a(1 + (2.7h^{\mathrm{up}}/W)^{0.77})^{1.3}$ is (a good fit of) the disconnected infiltration rate without clogging (Swamee et al., 2000). The numerical integration of Eq. (3) coupled with Eq. (4) and Eq. (5) is referred to as the exact stream length $L_c$. In the absence of clogging the notation $L_u$ is used.

It is shown in Paccolat and Battin (in preparation) that the solution to Eq. (4) is well approximated by the linear expression

$$q_c(h^{\mathrm{up}}) \approx q_0 + h^{\mathrm{up}}/R_c, \tag{6}$$

where $R_c = w_c/K_c$ is the clogging resistance (related to the resistivity by $R_c = r_c \nu/g$, with $g$ the gravitational acceleration and $\nu$ the water kinematic viscosity) and $q_0 = q_c(0)$ is the channel infiltrability. This work also reveals asymptotic solutions associated to three clogging categories, namely

$$q_0 \rightarrow \begin{cases} K_a, & \text{for } R_c \ll R_- & \text{(negligible clogging)} \\ K_a \left( R_c/R_- \right)^{-\xi}, & \text{for } R_- \ll R_c \ll R_+ & \text{(soft clogging)}, \\ K_c, & \text{for } R_c \gg R_+ & \text{(hard clogging)} \end{cases} \tag{7}$$

with $\xi = \frac{5n-1}{5n+1}$. The transition resistances $R_- = h_g/K_a$ and $R_+ = R_-(K_a/K_c)^{(5n+1)/(5n-1)}$ delimit the three regimes: increasing the clogging resistance, the infiltration is first controlled by the underlying sediments (negligible clogging), then depends on both layers (soft clogging) and is finally controlled by the clogging layer only (hard clogging). Using the approximate Eq. (6) and assuming wide channel conditions ($W \gg h^{\mathrm{up}}$), Eq. (3) integrates to the approximate clogged stream length (see Appendix A)

$$L_c^{\mathrm{app}} = \frac{Q_{\mathrm{in}}}{q_0} g_c \left( \frac{h_{\mathrm{in}}}{q_0 R_c} \right). \tag{8}$$

The hypergeometric function $g_c$ decreases from 1 to 0.5 for arguments growing from 0 to 2. By definition, this solution only applies if $h_{\mathrm{in}} < h_c$[1]. Similarly, assuming wide channel conditions ($W \gg h^{\mathrm{up}}$) for the unclogged configuration, Eq. (3) integrates

---

[1]The domain of applicability is actually slightly reduced due to the linear infiltration approximation and is found from $q_0 + h_{\mathrm{in}}/R_c = K_a$.

to the approximate unclogged stream length (see Appendix A)

$$L_u^{\text{app}} = \frac{Q_{\text{in}}}{K_a} \, g_u \left( \frac{h_{\text{in}}}{W} \right),$$ (9)

where the hypergeometric function $g_u$ decreases from 1 to 0.5 for arguments growing from 0 to 0.5.

### 2.2.2    Elongation factor

Clogging may increase stream length by reducing infiltration. This effect is quantified by the *elongation factor*, the ratio of
clogged to unclogged stream length, i.e., $\text{EF} = L_c/L_u$. At the scale of the terrace, we introduce the multi-stream elongation factor

$$\text{EF}_m = \frac{\sum_{i=1}^{m} L_c^{(i)}}{\sum_{i=1}^{m} L_u^{(i)}},$$ (10)

where $m$ is the number of streams draining the terrace. $L_c^{(i)}$ and $L_u^{(i)}$ are the clogged and unclogged stream lengths associated to the $i$th inlet of ponding depth $h_{\text{in}}^{(i)}$. Confluent and divergent stream branching will reduce, respectively, increase the overall
streams length (see Appendix A). However, because clogging only slightly alters ponding depth, the latter corrections almost cancel out from the elongation factor and are thus discarded.

### 2.2.3    Input parameter sets

To account for the variability of spatiotemporal factors such as inflows, pedology and topography, stream properties (i.e., $W$, $s$, $n_M$ and $h_{\text{in}}$) are drawn from independent distributions introduced in the previous subsubsections. Clogging properties (i.e.,
$w_c$ and $K_c$) are drawn from independent distributions characterized from experiments discussed in Section 3. The sediment hydraulic properties (i.e., $K_a$, $h_g$ and $n$) are drawn from joint distributions, either the sand or the floodplain distribution introduced previously. Altogether, the distribution of the nine input parameters is referred to as the sand parameter distribution (SPD) or the floodplain parameter distribution (FPD), depending on the considered distribution of sediment hydraulic properties, as summarized in Fig. 4a. Note that despite the randomness of input parameters each stream is described by a fixed set of
parameters (uniform conditions).

## 3    Experimental results

Similar microbial mat evolution was observed during the five flume experiments (Fig. 2a): sediment interstices were first filled within a few days before the mat grew homogeneously over the sediment bed to reach a maximal thickness of roughly $5\,\text{mm}$ within 15 days. Because of technical issues, *r1* was stopped during the growth phase. Mat permeability remained within a
relatively small range, well captured by a 10-fold window ($6 \times 10^{-6}\,\text{mm}^2 < k_c < 6 \times 10^{-5}\,\text{mm}^2$), with an inter-run variability comparable to intra-run variability. The initial reduction of permeability suggests that the mat grew denser while colonizing the sediments, to stabilize later to a more constant structure. Intense phototrophic activity resulted in the formation of oxygen

bubbles within and on top of the microbial mat forming vertical, tube-like connections through the mat. Particularly large bubbles (up to a few millimeters) were observed during *v1*, *v2* and *r3*. The buoyancy force resulted in large bubbles escaping

and perforating the mat on day 5 of *v2*, which corresponds to the observed sudden reduction of permeability. Note that only measurements taken at dawn were considered to discard diel permeability changes. Roncoroni et al. (2023a) conducted a similar experiment with faster flow and coarser sediments. The authors report a similar microbial mat growth reaching a maximal thickness of $8(3)\,\mathrm{mm}$ within 10 days. With a simplified flow path calculation, we estimated, from the reported infiltration rates, that the clogging layer permeability was $3.2(1.3) \times 10^{-7}\,\mathrm{mm}^2$ (see Appendix B).

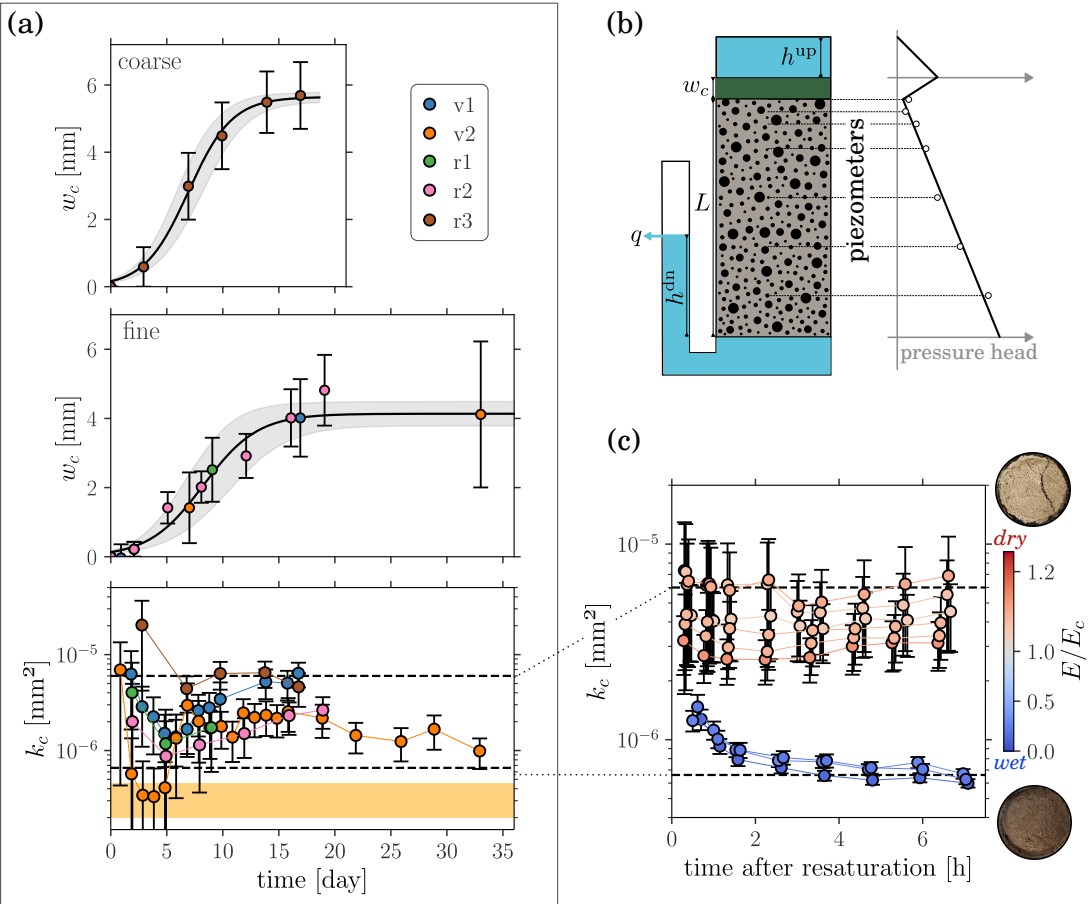

**Figure 2.** (a) Evolution of the mat thickness on coarse (top) and fine (middle) sediments, and of the mat permeability (bottom), during all flume experiments. The solid line and the shaded area respectively correspond to the best fit of the logistic function and the uncertainty arising from varying the fit parameters within two standard deviations. The dashed lines delimit the representative 10-fold window. The yellow band corresponds to the range of values obtained from Roncoroni et al. (2023a). (b) Sediment box and pressure head profile. (c) Evolution of the mat permeability after desiccation. Colors indicate mat dryness before resaturation.

Permeability measurements rely on sediment uniformity, which is corroborated by the algorithm: despite a reduction in sediment permeability (by 10 to 30 % for fine and 80 % for coarse sediments), the goodness-of-fit improved over time for all experimental runs (see SI Section 2). Permeability reduction is explained by the formation of trapped bubbles in the sediment matrix due to outgassing from air saturated water. Also, we were unable to quantify organic matter (as ash free dry mass) within the cores sampled at the end of *r2* and *r3*.


    Dried mats were characterized by the evaporation depth (integrated evaporation rate), $E$, at withdrawal normalized by the evaporation depth, $E_c$, at the transition to the second phase of evaporation (when no capillary water is left in the mat). The permeability of dry mats ($E > E_c$) was significantly (Welch's t-test: $p = 0.00$, $t = 4.29$, $df = 18$) larger than the one of wet mats ($E < E_c$), but all values remained within the 10-fold window obtained from the flume experiments (Fig. 2c). Crack
formation in dry mats probably explain the permeability rise. Permeability reduction of wet mats following re-saturation may be attributed to remobilized particulates.

## 4   Terrace model results

### 4.1   Stream length

    Stream length, $L$, is defined in Section 2.2 as the distance over which water flows before it infiltrates underground. Figure 3
illustrates how $L$ increases as clogging reduces infiltration for different inlet depths and clogging hydraulic conductivities. The hydraulic properties of the terrace sediments matches the mean of the floodplain distribution (i.e., $K_a = 1.9 \times 10^{-3}\,\mathrm{m\,s^{-1}}$, $h_g = 0.14\,\mathrm{m}$ and $n = 2.74$), while the channel slope, width and roughness are respectively set to $s = 1 \times 10^{-3}$, $W = 1\,\mathrm{m}$ and $n_M = 0.022\,\mathrm{s\,m^{-1/3}}$. For all combinations, three phases can be identified. When clogging is too small to cause unsaturated condition in the underlying sediments ($h_{\mathrm{in}} > h_c$), the stream length remains close to its unclogged reference value. For intermediate
resistances, it scales as $L \sim R_c^\xi$, with $\xi = (5n - 1)/(5n + 1)$, as predicted by the approximate solution for soft clogging (Eq. (8)). Finally, for resistances larger than $R_+$ (hard clogging), the stream length converges to a maximal value which depends on $K_c$.

### 4.2   Single-stream elongation factor

    To evaluate the extension of terrace streams due to mat induced clogging, the single stream (i.e., $m = 1$) elongation factor (Eq.
(10)) was computed for $N = 10,000$ terrace configurations (0.03 (SPD) and 0.9 (FPD) margin of error on the mean elongation factor at 95 % confidence interval) drawn from each input parameter distribution introduced in Section 2.2, i.e., SPD and FPD. The results from the previous section suggest the following bounds on the mat thickness and its hydraulic conductivity: $4\,\mathrm{mm} < w_c < 10\,\mathrm{mm}$ and $1.3 \times 10^{-6}\,\mathrm{m\,s^{-1}} < K_c < 4 \times 10^{-5}\,\mathrm{m\,s^{-1}}$. Values were respectively drawn from the uniform and log-uniform distributions. The input distributions are summarized in Fig. 4a and the output EF distributions are shown on Fig.
4b. For the SPD, the median (IQR) is 1.3 (1.1–2.2) and only 35 % of the configurations are unsaturated. For the FPD, the

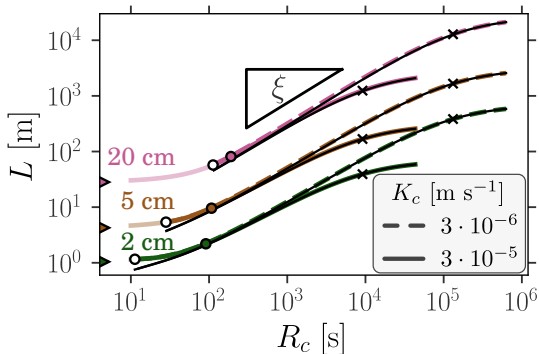

**Figure 3.** Stream length as function of clogging resistance for different inlet depth (colors) and clogging hydraulic conductivities (line styles). The exact and approximate clogged channel length $L_c$ and $L_c^{\text{app}}$ are respectively represented by the thick colored lines and the thin black lines. The saturation transition, $h_{\text{in}} = h_c$, is marked by white dots and curves are shaded in the saturated region. The color triangles correspond to the values associated to the unclogged limit. The colored dots indicate the limit of application of the approximate formula. The crosses mark the transition point $R_+$. The predicted scaling is outlined by the triangle slope ($\xi \approx 0.86$).

median (IQR) is 7.2 (2.4–24.2) and $85\,\%$ of the configurations are unsaturated. The fraction of streams shorter than a given (cutoff) terrace size is represented in Fig. 4c for both input distributions. Considering finite terrace sizes, the EF distributions are shifted toward smaller values, as exemplified on Fig. 4b.

The exact (numerical integration) and approximate (analytical) stream length solutions are compared in Fig. 3. The approximation only applies to unsaturated conditions as it relies on the approximate linear formula for unsaturated infiltration. Using the analytical solutions, the elongation factor takes the simple expression

$$\text{EF} \approx \frac{K_a}{q_0} \frac{g_c(h_{\text{in}}/(q_0 R_c))}{g_u(h_{\text{in}}/W)}, \tag{11}$$

which is compared to the exact solution in Fig. 5. Consistently, it is only reliable for unsaturated conditions. In particular, the relative error vanishes with increasing EF values, since the linear infiltration approximation improves for larger clogging resistances (Paccolat and Battin, in preparation).

For both input parameter distributions, the two most influential factors are the sediment hydraulic properties (grouped as a single factor due to the variable dependence) and the clogging hydraulic conductivity: for the FPD (SPD), the sediments explain $43.5\,\%$ ($16.2\,\%$), $K_c$ $17.6\,\%$ ($50.4\,\%$) and the interaction between both factors $30.7\,\%$ ($16.4\,\%$) of the EF variance. First- and total-order Sobol indices of the different factors are shown on Fig. 4d. The inlet depth strongly influences stream length (see Fig. 3), but its effect on the elongation factor is highly reduced as it almost factorizes out. Equation (11) reveals that $h_{\text{in}}$ only impacts EF through the $\mathcal{O}(1)$ term $g_c/g_u$. In particular, the sign and magnitude of the effect depend on the stream width and clogging. Also, the elongation factor is independent of the slope and the roughness coefficient since they factorize out from the stream length integral (i.e., $L \sim \sqrt{s}/n_M$).

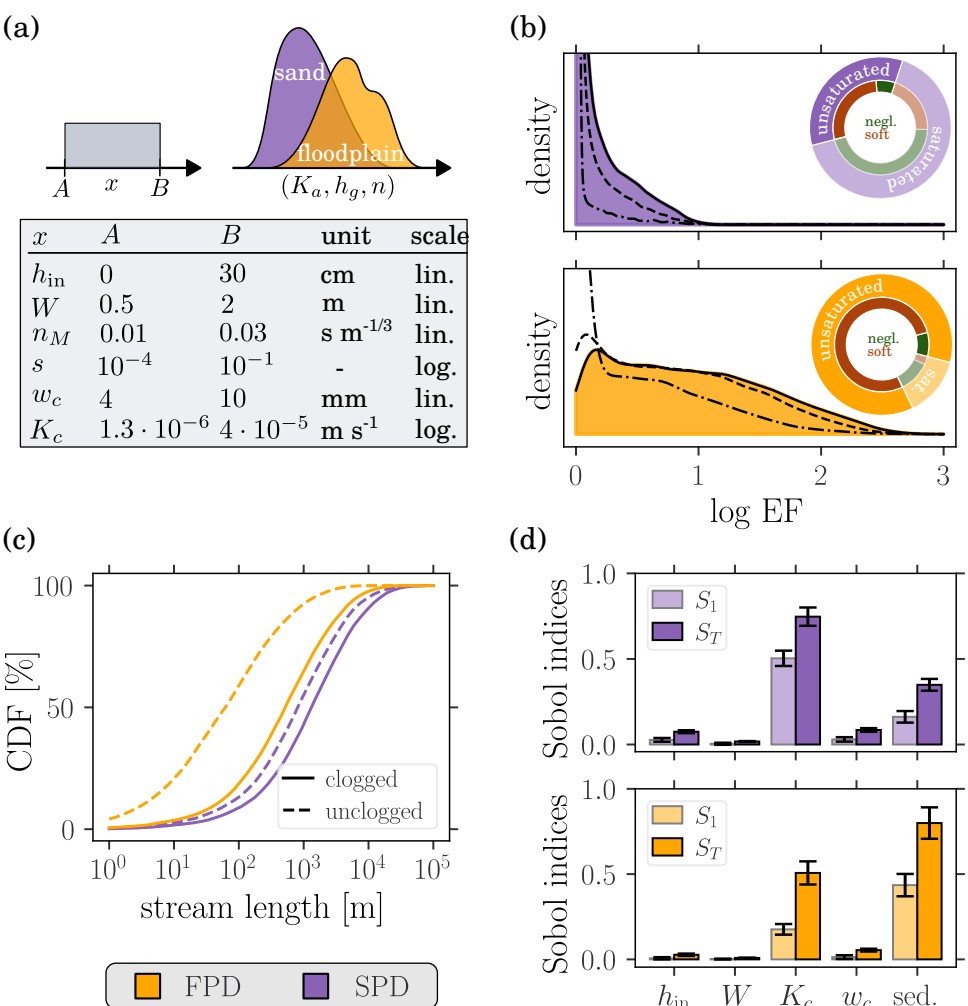

**Figure 4.** Single stream elongation factor (EF) for both SPD (violet) and FPD (yellow). (a) Input distributions. Bounds on the (log)-uniform distributions of the stream parameters and joint distributions of the sediment hydraulic properties. (b) Distribution of the elongation factor. Solid lines correspond to infinite terraces, while dashed and dash-dotted lines correspond to finite terraces (resp. of length $1000\,\mathrm{m}$ and $100\,\mathrm{m}$). The pie charts indicate the prevalence of unsaturated configurations, as well as of the different clogging categories. (c) Cumulative distribution function of stream lengths. (d) Sensitivity analysis ($2^{12}$ samples). $S_1$ and $S_T$ are respectively the first and total order Sobol indices. Error bars display the $95\,\%$ confidence intervals.

### 4.3 Multi-stream elongation factor

The multi-stream system also exhibits the three phase behavior as illustrated for the 10-stream elongation factor in Fig. 6. However, the soft clogging regime no longer converges to a power law. Indeed, in general $\mathrm{EF}_m \sim \sum_{i=1}^{m} A_i R_c^{\xi_i}$, with $A_i$ random variables depending on the stream, clogging and sediment properties. However, due to the small support of $\xi$, the apparent

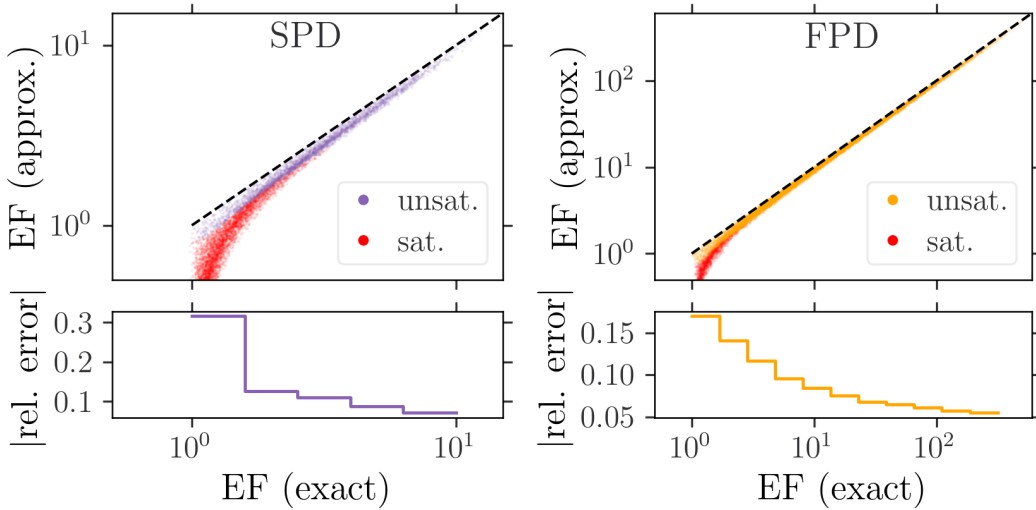

**Figure 5.** Validation of the approximate (analytical) EF expression for both SPD (violet) and FPD (yellow). Dashed lines indicate the identity function. Saturated configurations (red dots) are discarded in the relative error computation.

scaling EF $\sim R_c^{\xi_m}$ is observed. The distribution of $\xi_m$ is different from the one of $\xi$ because of two reasons. First, only a fraction of the considered configurations falls into the soft clogging category (see Fig. 4b). And second, even if considering larger clogging resistances corresponding to the maximal logarithmic slope (gray distributions in Fig. 6), the asymptotic behavior is not fully established because $K_a/K_c$ is not diverging, yielding smaller effective exponents. Nonetheless, with $81\,\%$ of soft clogging configurations and a $K_a/K_c$ median (IQR) of 126 (39–394), the FPD results in a fairly good scaling and exponents mostly between 2/3 and 1. Besides altering the scaling behavior, the presence of multiple streams averages out the elongation factor. It follows from Jensen's inequality that $\mathbb{E}[EF_m] \geq \mathbb{E}[L_c]/\mathbb{E}[L_u] = \mathbb{E}[EF_\infty]$. Numerical computations (data not shown) indicate further that $\mathbb{E}[EF_m] > \mathbb{E}[EF_{m+1}]$. In other words, the clogging effect decreases with the number of inlets in the stream network.

### 4.4 Model assumptions

The above results rely on the assumption of stream disconnection and steady-state. Rough estimates of the sum of the mound and capillary zone heights, discussed in Appendix C, indicate a median (IQR) of 15 (10–26) cm for the FDP and 18 (26–37) cm for the SDP. Similarly, the transient time (surface front propagation plus transient infiltration time) is quantified in Appendix C, yielding a median (IQR) of 1.1 (0.7–1.8) h for the FPD and 0.4 (0.2–0.8) h for the SPD.

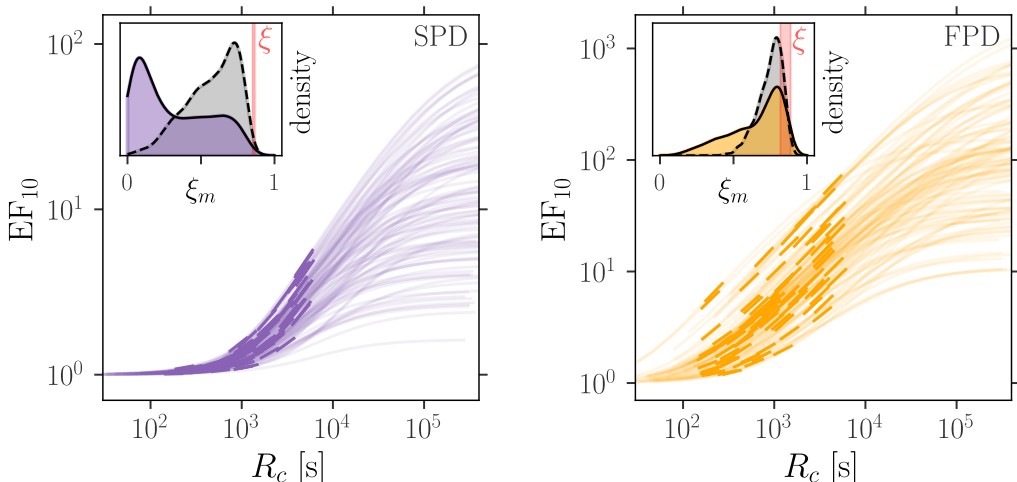

**Figure 6.** Elongation factor for 1000 multi-streams systems ($m = 10$) as function of the clogging resistance. All parameters are drawn from the terrace model input distributions (Fig. 4a), but $w_c$ which scans a wider range. The thick intervals indicate the mat thickness values from the terrace model. The slope distributions are highlighted in the inset plots (max slope in gray and mean thick interval slope in color). The red band corresponds to the range of $\xi$ values from the considered sediment distribution.

## 5 Discussion and conclusion

Combining experimental observations with idealized modeling, we here show how microbially induced clogging contributes to the habitability of proglacial landscapes. While stream length elongation may be considerable (i.e., up to 100 fold), sedimentary

properties typically dictate the magnitude of this effect (i.e., geomorphic control). However, our work clearly indicates that microbially induced clogging can be ecologically relevant in proglacial landscapes – by retaining water available for pioneer and early successional plants growing adjacent to proglacial streams, and increasing habitat size and connectivity available for aquatic organisms. While the importance of sediment biostabilization in intertidal zones and fluvial systems has already been recognized (Paterson et al., 2018), the relevance of this impermeabilization process during early successional stages of

proglacial ecosystems remained largely overlooked.

### 5.1 Microbial mat permeability

Our experimental results show that the microbial mat permeability varies between $6 \times 10^{-6}\,\mathrm{mm^2}$ and $6 \times 10^{-5}\,\mathrm{mm^2}$ independent of sediment type and experiment context (e.g., water source, meteorological conditions), and compares well to values estimated from Roncoroni et al. (2023a), i.e., $3.2(1.3) \times 10^{-7}\,\mathrm{mm^2}$. These experiments provide a first sensible range of the

permeability of microbial mats in streams, which improves on the eight orders of magnitude variability reported from model biofilm systems (Kurz et al., 2023). These values are however limited to artificial setups and field measurements are critically

lacking. Our results on the resilience of the permeability of microbial mats to desiccation corroborate infiltration observations from Roncoroni et al. (2023a). Also the relatively short growth timescales (up to 15 days) and widely reported cases of fast mat recovery following droughts (e.g., Hawes and Howard-Williams, 1998; Romaní et al., 2013; Touchette et al., 2025) support the first order approximation of constant mat properties used in the terrace model.

## 5.2 Stream elongation

It was postulated that microbial mat clogging can increase surface water availability (Miller and Lane, 2019; Roncoroni et al., 2019). The terrace model provides a simplified description of this effect. Clogging is shown to significantly increase stream length if large enough to unsaturate the underlying sediments, i.e., if $h_{\text{in}} < w_c(K_a/K_c - 1)$ (Brunner et al., 2009), which generally applies to the FPD but not to the SPD. For such configurations, the elongation factor is well approximated by the analytical solution (Eq. (11)), which highlights the effect of the different stream and clogging properties. In case of soft clogging (prevalent for both parameter distributions), the elongation factor (but also the stream length) scales with the clogging resistance, i.e., EF $\sim (w_c/K_c)^\xi$, where $\xi = (5n-1)/(5n+1) \in [2/3, 1]$ and $n$ increases with the uniformity of the sediment sizes. For a network of $m$ streams of different properties, an apparent scaling subsists (due to the small exponent range) with an effective exponent $\xi_m < \xi$. This observation offers a rule of thumb to predict surface water cover response to clogging: for a $x$-fold increase of the clogging resistance, the overall length of the streams increases by $x^{\xi_m}$, which may help anticipate expansion–contraction dynamics.

## 5.3 Uncertainty and sensitivity

Because of the inherent heterogeneity of the physical parameters involved in the terrace model (i.e., $h_{\text{in}}$, $W$, $s$, $n_M$, $K_a$, $h_g$, $n$, $w_c$ and $K_c$) and their values being poorly constrained from field observations, the uncertainty analysis indicates elongation factors ranging from one (no effect) to more than one hundred. The wider range and larger values found with the FPD than with the SPD are explained from the hydraulic conductivities being larger for the former distribution. More precisely, they are sufficiently large to trigger unsaturation for most configurations, resulting in large clogging effect, while this is not the case for the SPD. Fixing the input parameter distribution, the sensitivity analysis showed that the sediment hydraulic properties are the first (FPD) or second (SPD) most influential parameters. Gathering both distributions in the same sensitivity analysis (e.g. using a random trigger to select the distribution (Puy et al., 2020)) would undoubtedly result in a strong dominance of the latter parameters on the EF variability. Reducing uncertainty on the sediment hydraulic properties is consequently of prime importance to improve clogging effect estimates.

Because the stream length is proportional to $\sqrt{s}/n_M$, the elongation factor is independent of these two parameters. In reality, streambed roughness is altered by microbial mat growth such that EF should be corrected by a factor $n_{M,u}/n_{M,c}$, where the indices refer to unclogged and clogged conditions, respectively. For reduced roughness, as observed by Roncoroni et al. (2023a), the clogging effect would hence be accentuated.

Intermittent network dynamics are observed in glacial forefields (Paillex et al., 2020; Malard et al., 1999) in response to the multiple water inputs (i.e., from kryal, krenal and stormflow sources) and variable landform transmissivities. Re-

solving this complex behavior is, however, not necessary to study the effect of clogging since the elongation factor is only slightly influenced by the inflow. Nonetheless, the effect is slightly larger during low flow periods as quantified by the ratio $\text{EF}|_{h_{\text{in}}=20\,\text{cm}}/\text{EF}|_{h_{\text{in}}=2\,\text{cm}}$, whose median (IQR) is 0.7 (0.8–1.0) for the SPD and 0.7 (0.8–0.9) for the FPD (data not shown).

## 5.4   Validity of the terrace model assumptions

The terrace model relies on several idealizations (Section 2.2). First, groundwater disconnection is motivated by the relative
elevation of the terrace with respect to the main glacier-fed streams, which control the position of the water table, as well as diverse field observations (e.g. Miller and Lane, 2019; Malard et al., 1999; Siegfried et al., 2023). Estimates of vadose zone and water mounding, for predicted infiltration rates, in Section 4.4, corroborate further this intuition – terraces elevation of a few decimeters would be sufficient for disconnection. However, these values strongly depend on the arbitrarily chosen terrace geometry and must be taken cautiously. Second, the steady-state assumption requires stream response to be fast compared to
inflow variations. Time to complete stream inundation is estimated around one hour in Section 4.4 for the parameters used in the study, such that response to inflow variations – even faster due to the smaller amount of water involved – may safely be assumed instantaneous in front of diurnal and seasonal timescales. Finally, parameters describing flow and infiltration (i.e., $w_c$, $K_c$, $K_a$, $h_g$, $n$, $n_M$, $s$ and $W$) are assumed homogeneous along each stream. This is an arguably necessary first order approximation in front of our ignorance regarding spatial distributions. Also it permits an explicit formulation of stream length (i.e., Eq. (3))
from the conservation equation, $dQ/dx = -q(h^{\text{up}})$, by separation of variables. While specific studies could address the effect of in-stream heterogeneity, it is generally expected that the variability of stream lengths will be reduced if part or all of the between stream variability is present at the single stream scale.

## 5.5   Conclusion

Although highly idealized, the terrace model provides a mechanistic description of stream elongation, which refines our view
on the bio-engineering potential of microbial mats and shows their transitional role linking the pioneer and the biogeomorphic ecosystem phases. The extent to which primary succession benefits from microbial mats remains, however, poorly quantified, in view of the variability of parameters and processes involved. Nevertheless, the terrace model indicates that for given stream conditions, impermeabilization is a relevant mechanism if (1) clogging is sufficient to unsaturate the underlying sediments, i.e., $h_{\text{in}} < w_c(K_a/K_c - 1)$, and (2) the terrace is longer than the unclogged stream length, $L_u \approx Q_{\text{in}}/K_a\,g_u(h_{\text{in}}/W)$. Under
these conditions, the effect can be quantified by the clogged stream length formula, $L_c \approx Q_{\text{in}}/q_0\,g_c(h_{\text{in}}/q_0 R_c)$. Resolving the dynamics of streams draining proglacial terraces would require more realistic models accounting for additional processes (e.g., mat dynamics, coupling with water flow and nearby vegetation). However, model calibration is delusive with the current state of knowledge. Site specific, data-driven studies could provide field evidence for our predictions and set premises for further model development. In particular, drone-based imagery has shown promising results to simultaneously map flow and biofilms
(Roncoroni et al., 2023b).

*Code and data availability.* Codes and data are freely accessible from the following GitLab repositories: https://gitlab.com/jonas.paccolat/ flume-monitor (flume experiments), https://gitlab.com/jonas.paccolat/matcores (desiccation experiments) and https://gitlab.com/jonas.paccolat/ terrace-model (terrace model simulations). All figures can be generated from Jupyter notebooks in the respective repositories.

## Appendix A:  Approximate stream length and branching

**A1    Stream length integral**

The stream length integral (Eq. (3)) can be solved analytically for the clogged unsaturated ($h_{\text{in}} < h_c$) and the unclogged configurations under the wide channel assumption ($W \ll h_{\text{in}}$). In both cases, we make use of the equality

$$
{}_2F_1(1, b, b+1, z) = \int\limits_0^1 \frac{du}{1 - zu^{1/b}}, \tag{A1}
$$

which is a particular case of the Euler integral expression of the hypergeometric function. For clogged unsaturated profiles,

injecting the approximate linear infiltration equation $q_c(h^{\text{up}}) \approx q_0 + h^{\text{up}}/R_c$ into the stream length integral yields

$$
\begin{aligned}
L_c^{\text{app}} &= \int\limits_0^{Q_{\text{in}}} \frac{dQ}{q_0 + h^{\text{up}}(Q)/R_c} \\
&= \frac{1}{q_0} \int\limits_0^{Q_{\text{in}}} \frac{dQ}{1 + \frac{\left(n_M Q/\sqrt{s}\right)^{3/5}}{h^{\star}}} \\
&= \frac{Q_{\text{in}}}{q_0} \int\limits_0^1 \frac{du}{1 + \frac{h_{\text{in}}}{h^{\star}} u^{3/5}} \\
&= \frac{Q_{\text{in}}}{q_0} {}_2F_1(1, 5/3, 8/3, -h_{\text{in}}/h^{\star}) = \frac{Q_{\text{in}}}{q_0} g_c(h_{\text{in}}/h^{\star}),
\end{aligned}
$$

where $h^{\star} = q_0 R_c$ and $h_{\text{in}} = (n_M Q_{\text{in}}/\sqrt{s})^{3/5}$. The wide channel assumption was used to simplify the Manning formula in the second line and the third line follows from the change of variable $u = Q/Q_{\text{in}}$. It is shown in Swamee et al. (2000) that seepage from rectangular channels into unconfined aquifer is well fitted by the function $q_u(h^{\text{up}}) = K_a \left[1 + (a' h^{\text{up}}/W)^{0.77}\right]^{1.3} \approx K_a \left[1 + a (h^{\text{up}}/W)^{0.77}\right]$, where the approximation follows from the wide channel assumption and $a = 1.3a'^{0.77} \approx 2.79$, with $a' = \pi(4 - \pi)$. This expression can be used to solve the unclogged channel length integral, by following the same steps as in

 the previous integral, namely

$$L_u^{\text{app}} = \frac{1}{K_a} \int_0^{Q_{\text{in}}} \frac{dQ}{1 + a(h^{\text{up}}(Q)/W)^{0.77}}$$

$$= \frac{Q_{\text{in}}}{K_a} \int_0^1 \frac{du}{1 + a\left(\frac{h_{\text{in}}}{W}\right)^{0.77} u^{0.462}}$$

$$= \frac{Q_{\text{in}}}{K_a} {}_2F_1\left(1, 2.164, 3.164, -a(h_{\text{in}}/W)^{0.77}\right) = \frac{Q_{\text{in}}}{K_a} g_u(h_{\text{in}}/W).$$

The functions $g_c$ and $g_u$ are plotted in Fig. A1a.

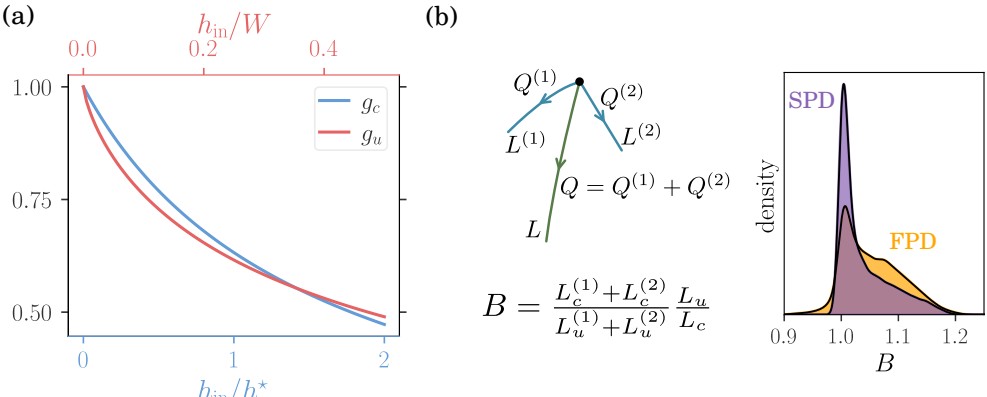

**Figure A1.** (a) Hypergeometric functions associated to the clogged (blue) and unclogged (red) streambeds. (b) Construction of the branching ratio and its distribution for both sand and floodplain parameter sets.

## A2 Branching

For a given set of inflows generating a stream network, the total network length depends on the presence and positions of stream branchings. If two streams of respective surface flow $Q_1$ and $Q_2$ merge into a single stream of surface flow $Q = Q_1 + Q_2$, the length of the resulting stream $L$ is smaller than the sum of the two independent streams ($L_1$ and $L_2$). To estimate this effect on the elongation factor, we define the (two streams) branching ratio as the ratio between the elongation factor with and without branching, namely

$$B = \frac{L_c^{(1)} + L_c^{(2)}}{L_u^{(1)} + L_u^{(2)}} \times \frac{L_u}{L_c} \approx \frac{Q^{(1)} g_c^{(1)} + Q^{(2)} g_c^{(2)}}{Q^{(1)} g_u^{(1)} + Q^{(2)} g_u^{(2)}} \times \frac{g_u}{g_c}, \tag{A2}$$

where the approximation only applies to unsaturated conditions and $g_c = g_c(h^{\text{up}}/h^\star)$, $g_u = g_u(h^{\text{up}}/W)$, $g_c^{(i)} = g_c(h_i^{\text{up}}/h^\star)$ and $g_u^{(i)} = g_u(h_i^{\text{up}}/W)$ for $i = 1, 2$. The ponding depth $h^{\text{up}}$ and $h_i^{\text{up}}$ are computed with the Manning equation and the respective

inflows $Q$ and $Q_i$. Considering both parameter distributions, the branching ratio essentially remains between 1 and 1.1 (Fig.
A1b). A stream may also split in two, resulting in a increased total length. In that case the branching ratio is inverted (i.e., $B^{-1}$) such that the branching effect is negligible at the multi-stream scale.

## Appendix B:  Estimation of microbial mat permeability from Roncoroni et al. (2023a)

During the flume experiment carried out by Roncoroni et al. (2023a), the infiltration flux dropped from $Q_i = 0.18(1)\,\mathrm{L\,s^{-1}}$ to $Q_f = 0.025(5)\,\mathrm{L\,s^{-1}}$ as a consequence of microbial mat growth. The sediment chamber length, height and width were respectively $L_x = 2\,\mathrm{m}$, $L_z = 7\,\mathrm{cm}$ and $W = 30\,\mathrm{cm}$, while the ponding depth was $h^{\mathrm{up}} = 11.5\,\mathrm{cm}$. The infiltrated water was drained through three pipes of diameter $D_{\mathrm{pipe}} = 7\,\mathrm{mm}$ and unknown length $L_{\mathrm{pipe}} = 10$ to $20\,\mathrm{cm}$ (range estimated by the experimental designer upon request). In order to estimate the sediment permeability, $k_s$, and the clogging resistivity, $r_c$, we first compute the pressure at the pipe entrance according to the Darcy-Weisenbach equation

$$h_{\mathrm{out}} = \frac{L_{\mathrm{pipe}}}{D_{\mathrm{pipe}}} \frac{f_D(Re)\,u^2}{2g}, \tag{B1}$$

where $u$ is the mean flow through each pipe obtained from $Q_{\mathrm{pipe}} = Q/3 = \pi(D/2)^2 u$, $\rho$ is the water density and $f_D$ is the Darcy friction factor. The Reynolds numbers of the pipe flows are respectively $Re_i = 7200(400)$ and $Re_f = 1000(200)$ for the initial and final rates. The friction factors $f_D(Re_i)$ and $f_D(Re_f)$ are thus respectively computed from the turbulent smooth-pipe and the laminar expressions. We then compute the total flux $Q$ according to the chamber dimensions and boundary conditions. It is given by the integral

$$Q = W \int\limits_0^{L_x} q(x)dx, \tag{B2}$$

where $q(x)$ is the flow rate along the streamline starting a distance $x$ from the chamber downstream end. We solve this integral by assuming that the streamlines follow a right-angle path of length $L = x + L_z + w_c$, where $w_c$ is the clogging layer thickness (Fig. B1a). Consequently the flow rates are given by

$$q(x) = K \frac{h^{\mathrm{up}} - h_{\mathrm{out}} + L_z}{x + L_z + w_c} = K_s \frac{h^{\mathrm{up}} - h_{\mathrm{out}} + L_z}{x + L_z + r_c K_s}, \tag{B3}$$

where $K_s = g/\nu k_s$ is the sediment hydraulic conductivity and $K$ is the average hydraulic conductivity, which are related by $K/(x + L_z + w_c) = K_s/(x + L_z) + r_c$. The above integral hence yields

$$Q(k_s, r_c) \approx \frac{gbk_s}{\nu} (h^{\mathrm{up}} - h_{\mathrm{out}} + L_z) \log\left(1 + \frac{L_x}{L_z + k_s r_c}\right). \tag{B4}$$

The sediment permeability is found by solving this equation for the initial condition, i.e., $Q(k_s, 0) = Q_i$. The clogging resistivity is then obtained by solving it for the final condition, i.e., $Q(k_s, r_c) = Q_f$. The results depend on the unknown pipe length $L_{\mathrm{pipe}}$. Varying it from 10 to 20 cm results in a variability similar to the one arising from the uncertainty on the initial and final fluxes as shown in Fig. B1b. Altogether we find $k_s = 3.0(6) \times 10^{-4}\,\mathrm{mm^2}$ and $r_c = 2.5(3) \times 10^7\,\mathrm{mm^{-1}}$. With respect to the reported mat thickness of $w_c = 8(3)\,\mathrm{mm}$, we hence find $k_c = 3.2(1.3) \times 10^{-7}\,\mathrm{mm^2}$.

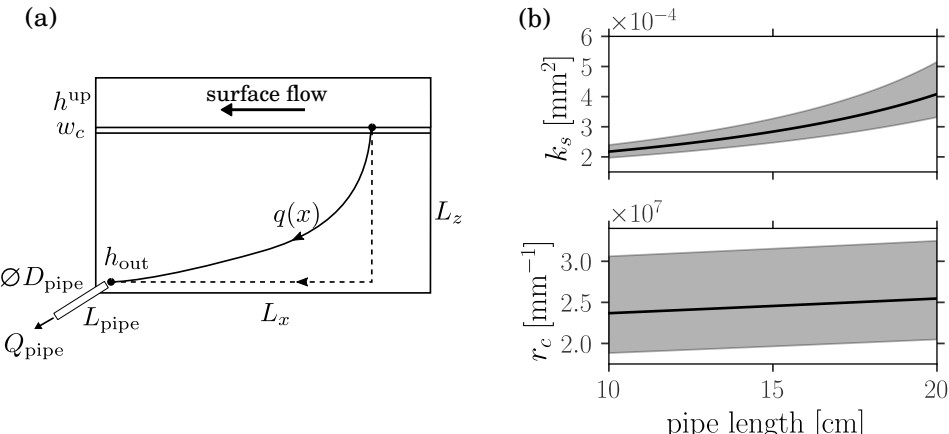

**Figure B1.** (a) Sketch of the flume experiment from Roncoroni et al. (2023a) illustrating the considered simplified flow trajectory. (b) Dependence of the sediment permeability and the clogging resistance on the unknown pipe length. The shaded areas correspond to the uncertainty propagated from the flow rate uncertainties.

### Appendix C: Assessment of the terrace model assumptions

#### C1 Groundwater mound

The water table rises below the stream to evacuate the flux laterally. For a two-dimensional symmetrical configuration, the linearized Boussinesq equation predicts the following water table level (Brunner et al., 2009)

$$\Delta z_{\text{WT}}(x) = \frac{q}{2T}(WL_a - W^2 - x^2), \quad \text{for } 0 < x < \frac{W}{2}, \tag{C1}$$

where $q$ is the infiltration rate, $x$ is the lateral distance from the centre of the stream of width $W$ and $L_a$ is the distance to the reference water table level. Considering an aquifer of depth $B_a$ and transmissivity $T = K_a B_a$, the height of the saturated mound

below the stream is $M_{\text{sat}} = qW(L_a - W/4)/2K_a B_a$. On top of the mound, a capillary zone extends the groundwater zone of influence. It is estimated from the following linear approximation. At steady-state, the infiltration rate within the unsaturated zone is equal to the one at the water table, yielding

$$q = K_a(-h_I) = K_a \left(1 - \frac{\partial h}{\partial z}\Big|_{\text{WT}}\right), \tag{C2}$$

where $h_I < 0$ is the constant pressure head below the clogged streambed. Assuming the pressure gradient to remain constant

within the capillary zone, the size of the latter is approximated as

$$w_{\text{cz}} \approx \frac{-h_I}{\partial h/\partial z|_{\text{WT}}} = \frac{K_a^{-1}(q)}{1 - q/K_a}, \tag{C3}$$

where $K_a^{-1}(\cdot)$ is the inverse of $K_a(\cdot)$, such that $-h_I = K_a^{-1}(q)$. Considering $B_{aq} = 8\,\mathrm{m}$ and $L_{aq} = 15\,\mathrm{m}$, the distribution of $M_{sat}$ and $w_{cz}$ are shown in Fig. C1a for both parameter distributions (SPD and FPD). Note that only the unsaturated configurations (i.e., $85\,\%$ for the FPD and $35\,\%$ for the SPD) are considered. For the saturated configurations, an inverted water table is formed below the stream which may significantly increase the required water table depth for disconnection (Xie et al., 2014; Xian et al., 2017).

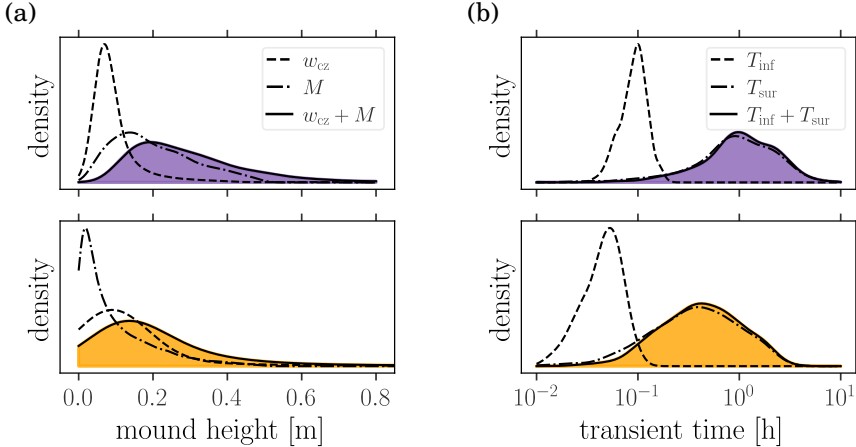

**Figure C1.** Distribution of (a) the saturated mound height and the capillary zone height, and (b) the surface and infiltration transient times obtained from the input parameter distributions of Fig. 4a (FPD in yellow and SPD in violet).

## C2 Transience

The problem of surface water propagation downward from a constant inflow $Q_{in}$ was studied by Turbak and Morel-Seytoux (1988). A power law rating curve is used which corresponds to the Manning formula in the large width limit, namely $Q = \sqrt{s}\,(h^{up})^{5/3}/n_M$. The position of the front at time $t$ and the time required for the front to reach the distance $x$ are respectively given by

$$x_F(t) = \frac{\sqrt{s}}{n_M I(t)} \left[ (h_{in})^{5/3} - \left( h_{in} - \frac{3}{5} I(t)t \right)^{5/3} \right],$$

$$t_F(x) = \frac{3}{5I(x)} \left[ h_{in} - \left( (h_{in})^{5/3} - \frac{n_M}{\sqrt{s}} x I(x) \right)^{3/5} \right],$$

where $I(t)$ and $I(x)$ correspond to the averaged infiltration rate up to time $t$, respectively up to distance $x$. Here we merely observe that when the front has propagated to its maximal distance $L$ after a time $T_{sur}$, one can set $x_F(T_{sur}) = L$, $t_F(L) = T_{sur}$ and $I(T_{sur}) = I(L) = I_{end}$ in the above two equations. Solving them yields

$$T_{sur} = \frac{5h_{in}}{3I_{end}} \quad \text{and} \quad L = \frac{\sqrt{s}\,(h_{in})^{5/3}}{n_M I_{end}},$$

which results in the relation $T_{\text{sur}} = 5n_M L/3\sqrt{s}(h_{\text{in}})^{2/3} = 5L/3u_{\text{in}}$. The transience time is increased by the infiltration re-laxation time $T_{\text{inf}}$ required for the infiltration rate to reach its maximal value. It is computed by solving the Richards equa-

tion for $h_{\text{in}} = 0$ (i.e., stream end) until the infiltration rate $q(t)$ stabilizes. The simulations is stopped when $\partial q/\partial t < \epsilon q$, with $\epsilon = 1 \times 10^{-3}\,\text{s}^{-1}$. The distribution of $T_{\text{sur}}$ and $T_{\text{inf}}$ are displayed in Fig. C1b for both parameter distributions.

*Author contributions.*  JP: conceptualization (lead), methodology, data curation, formal analysis, software, visualization, writing (original draft and review and editing (equal)). PA: conceptualization (support), supervision (support), writing (review and editing (equal)). SL: conceptualization (support), supervision (support), writing (review and editing (equal)). HP: conceptualization (support), supervision (support),

writing (review and editing (equal)). TB: conceptualization (support), supervision (lead), writing (review and editing (equal)).

*Competing interests.*  The authors declare that they have no conflict of interest.

*Acknowledgements.*  JP is very grateful to Nicola Neluigi, Charlotte Girard, Mathild Coyac and Hannes Peter for their help in the field. JP also thanks Laurent Morier and its team for the construction of the flume, as well as Simon Escalle and Robin Délèze for their technical expertise.

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
