# Peer review of "Microbial mats promote surface water retention in proglacial"

_EGUsphere, 2025_

## Author Response (AR1)

**Answer to referees**

We thank the referees for their insightful comments on our manuscript. Below, we provide point-to-point responses to all the comments. For clarity, the referees' comments are reproduced in blue, authors' replies are in black and modifications to the manuscript are in red.

**First referee**

1) The authors evaluated the effect of different sediments on mat permeability and EF. Have you also considered different types of biofilms? Did you quantify organic matter in the samples? The procedure for preparing the microbial mats is not entirely clear.

Microbial mats spontaneously developed over the sediments once flooded with stream water. Resolving the complex microbial community of the mat was beyond the scope of our study which was only concerned with its bulk physical properties. Organic matter was only quantified as ash free dry mass: 5 to 10 percent was found in the mat, but none was measurable within the sediments (below the mat). This observation is used in Line 199 to justify that permeability remained uniform within the sediments, a necessary condition for applying Equation 1.

To clarify this point Line 80 is modified as follows:
*As streamwater fed into the flume, microbial mats started colonising the sediment interface. Its thickness...*

2) Line 88: How did you account for water temperature? Could temperature affect the hydraulic conductivity of microbial mats? It seems no temperature measurements were included in the experiment.

Water temperature was monitored continuously (every minute) in the flume. The hydraulic conductivity, K, is proportional to the water kinematic viscosity, $\nu$, which decreases with temperature. Consequently, diurnal variations of temperature resulted in changes of infiltration rate, i.e. changes of K (of both the mat and the sediments). To account for this effect, the mat permeability -- which is independent on the fluid properties -- was considered instead, according to the relation k = K $\nu$ / g.

To clarify this point line 88 is modified as follows:
*Finally, to cancel out the effect of temperature on hydraulic conductivity, K -- water viscosity almost doubles in the range of temperature considered --, it is converted to permeability, k, according to the monitored temperature.*

3) Figure 4c: The black lines in the legend are confusing, as there is no solid black line visible in the figure. In addition, the figure is hard to read. If the solid lines indicate clogged conditions, why do they appear to show more drainage than the unclogged conditions?

We agree that the figure is confusing; solid and dashed lines are inverted and the overall description is misleading. Modifications are proposed below. First, some clarifications are needed: (1) The black line legend generically refers to both SPD (violet) and FPD (yellow).

(2) The vertical dashed and dash-dotted lines correspond to the cutoff used in Figure 4b. (3) The plotted (color) lines correspond to the cumulative distribution of stream lengths, or equivalently the fraction of streams smaller than a given terrace size.

The following modifications are proposed:
- Legend of Figure 4:
  *(b) Distribution of the elongation factor. Solid lines correspond to infinite terraces, while dashed and dash-dotted lines correspond to finite terraces (resp. of length 1000 m and 100 m). The pie charts indicate the prevalence of unsaturated configurations, as well as of the different clogging categories. (c) Cumulative distribution function of stream lengths.*
- Figure 4c:
  x-label: *stream length [m]*
  y-label: *CDF [%]*
  The vertical dashed and dash-dotted lines are removed.
  Solid and dashed legends are switched (corrected).
- Line 228:
  *Considering finite terrace sizes, the EF distributions are shifted toward smaller values, as exemplified on Figure 4b.*

4) Figure 4d: This figure is not cited in the main text and should be properly referenced.

Line 239 is modified as follows:
*for the FPD (SPD), the sediments explain 43.5 % (16.2 %), Kc 17.6 % (50.4 %) and the interaction between both factors 30.7 % (16.4 %) of the EF variance. First- and total-order Sobol indices of the different factors are shown on Figure 4d.*

5) I also encourage the authors to consider discussing a bit how microbial community structure and functional diversity may influence permeability and hydrological processes, as this could provide valuable ecological context for the findings.

Thank you for this comment. As already mentioned in the first point, the study of community structural and functional composition is well beyond the scope of this research. This would involve DNA extraction and sequencing, the information of which would not contribute to the mechanistic understanding of the processes that we focus on in this manuscript.

**Second referee**

1) The "terrace model" is a useful simplification for exploring the effects of microbial clogging on stream elongation. However, the assumptions of steady-state flow, groundwater disconnection, and homogeneous clogging may not hold across all proglacial settings. The authors briefly acknowledge these limitations in Section 5.4, but a more systematic discussion—perhaps in a dedicated subsection or figure caption—would strengthen the manuscript.

We are grateful for this comment. It is clear that our approach and framework serve as a model only, hence characterized by simplifications. To address this point we gather all comments regarding model assumptions in a new Section 5.4 and develop them.

*The terrace model relies on several idealizations (Section 2.2). First, groundwater disconnection is motivated by the relative elevation of the terrace with respect to the main glacier-fed streams, which control the position of the water table, as well as diverse field observations (e.g., Miller and Lane, 2019; Malard et al., 1999; Siegfried et al., 2023). Estimates of vadose zone and water mounding, for predicted infiltration rates, in Section 4.4 corroborate further this intuition -- terraces elevation of a few decimeters would be sufficient for disconnection. However, these values strongly depend on the arbitrarily chosen terrace geometry and must be taken cautiously. Second, the steady-state assumption requires stream response to be fast compared to inflow variations. Time to complete stream inundation is estimated around one hour in Section 4.4 for the parameters used in the study, such that response to inflow variations -- even faster due to the smaller amount of water involved -- may safely be assumed instantaneous in front of diurnal and seasonal timescales. Finally, parameters describing flow and infiltration (i.e., $w_c$, $K_c$, $K_a$, $h_g$, n, $n_M$, s and W) are assumed homogeneous along each stream. This is an arguably necessary first order approximation in front of our ignorance regarding spatial distributions. Also it permits an explicit formulation of stream length (Equation 3) from the conservation equation, i.e., dQ/dx = -q(h), by separation of variables. While specific studies could address the effect of in-stream heterogeneity, it is generally expected that the variability of stream lengths will be reduced if part or all of the between stream variability is present at the single stream scale.*

To avoid repetition, the following lines are removed: 127, 130-131 and 260.

2) Several figures, particularly Figure 4, contain overlapping distributions and small fonts that hinder readability. I recommend increasing font sizes, and potentially separating panels for clarity.

The following modifications are proposed:
Increase size of Figure 3, 5 and 6 and increase size of pie charts in Figure 4b. Also Figure 4c is lightened in agreement with response to the 3rd point of the first referee.

3) Although the authors note that the elongation factor is relatively insensitive to inflow variations, proglacial streams are inherently dynamic, with strong diurnal and seasonal discharge fluctuations. Given that clogging efficiency may vary with flow intensity (e.g., shear stress affecting mat integrity), could intermittent high flows reset or modulate the clogging process?

The terrace model shows that for a given microbial clogging, the elongation factor is relatively insensitive to inflow variations. However this is only an idealized description which neglects mat dynamics (i.e., growth, detachment, scouring, desiccation,...). Field observations of relatively stable mats (in the low flow terrace streams considered) nevertheless motivate this simplification; also the desiccation experiments support the hypothesis of constant clogging permeability. In addition, lack of understanding of mat dynamics, as well as coupling to flow dynamics and riparian vegetation, sets limits to what can be integrated in the model. We suggest site specific studies to investigate the dynamics, at the end of the manuscript, in the new section 5.5 (see last point):

*Resolving the dynamics of streams draining proglacial terraces would require more realistic models accounting for additional processes (e.g., mat dynamics, coupling with water flow and nearby vegetation). However, model calibration is delusive with the current state of knowledge. Site specific, data-driven studies could provide field evidence for our predictions and set premises for further model development. In particular, drone-based imagery has shown promising results to simultaneously map flow and biofilms (Roncoroni et al., 2023b).*

4) The paper assumes constant Manning roughness (nM) for both clogged and unclogged scenarios. However, microbial mat development is known to alter bed roughness—often reducing it by smoothing grain surfaces or increasing it via filamentous structures. Since roughness affects flow depth and velocity, ignoring its change may bias estimates of ponding depth and, consequently, infiltration rates. The authors should explicitly acknowledge this limitation and, if possible, provide a sensitivity estimate (even qualitatively) of how variable roughness might influence the elongation factor.

Bio-induced roughness variation is indeed a delicate point as it depends qualitatively on many poorly constrained factors. For this reason, we preferred to assume a null effect. Nevertheless, because the roughness coefficient factorizes out of the stream length integral, the effect of a quantified roughness change could easily be accounted for, as discussed in Section 5.3: "*Because the stream length is proportional to $\sqrt{s}/nM$, the elongation factor is independent of these two parameters. In reality, streambed roughness is altered by microbial mat growth such that EF should be corrected by a factor $nM,u/nM,c$, where the indices refer to unclogged and clogged conditions, respectively. For reduced roughness, as observed by Roncoroni et al. (2023a), the clogging effect would hence be accentuated.*" (Line 305)

5) The model predicts stream elongation ranging from none to 100-fold, a striking result. To enhance the practical impact of the study, the authors could discuss how these predictions might be tested in the field. For example, could drone-based thermal imaging, time-lapse photography, or tracer experiments be used to detect clogging-induced changes in surface water extent?

The last section is modified as follows to include suggestions for further research and better connect with the new section 5.4 (first point).

*5.5 Conclusion*
*Although highly idealized, the terrace model provides a mechanistic description of stream elongation which* refines our view on the bio-engineering potential of microbial mats and shows their transitional role linking the pioneer and the biogeomorphic ecosystem phases. *The extent to which primary succession benefits from microbial mats remains, however, poorly quantified, in view of the variability of parameters and processes involved. Nevertheless, the terrace model indicates that* for given stream conditions, impermeabilization is a relevant mechanism if (1) clogging is sufficient to unsaturate the underlying sediments, i.e., $hin < wc (Ka /Kc −1)$, and (2) the terrace is longer than the unclogged stream length, $Lu ≈ Qin /Ka gu (hin /W )$. Under these conditions, the effect can be quantified by the clogged stream length formula, $Lc ≈ Qin /q0 gc (hin /q0 Rc)$. *Resolving the dynamics of streams draining proglacial terraces would require more realistic models accounting for additional processes (e.g., mat dynamics, coupling with water flow and nearby vegetation). However, model calibration is delusive with the current state of*

*knowledge. Site specific, data-driven studies could provide field evidence for our predictions and set premises for further model development. In particular, drone-based imagery has shown promising results to simultaneously map flow and biofilms (Roncoroni et al., 2023b).*